# Mechanistic insights into alcohol-induced DNA crosslink repair by Slx4-Xpf-Ercc1 nuclease complex in the Fanconi anaemia pathway

Jana Havlikova[1,2], Milan Dejmek [1], Andrea Huskova[1], Anthony Allan[3], Evzen Boura [1], Radim Nencka [1] & Jan Silhan [1]✉

Alcohol is broken down in the body into acetaldehyde, a toxic chemical that can damage DNA by creating interstrand crosslinks (AA-ICL). These crosslinks block DNA replication and threaten the stability of the genome. A rare genetic disease, Fanconi anaemia (FA), is marked by extreme sensitivity to DNA crosslinking agents, including acetaldehyde. Although the Fanconi anaemia DNA repair pathway is known to fix this type of damage, exactly how it repairs acetaldehyde crosslinks is not yet understood. Here we show that the FA nuclease Slx4-Xpf-Ercc1 (SXE) plays a key role in the repair of AA-ICL. Using a DNA replication fork with site-specific AA-ICL, we show that SXE specifically excises this crosslink, highlighting its role in the repair of alcohol-induced DNA interstrand crosslinks. Moreover, SXE performs two precise incisions flanking the AA-ICL and can similarly repair a basic-site DNA interstrand crosslink. These results expand our understanding of how the FA pathway resolves alcohol-induced DNA damage. In addition, they suggest that SXE is a versatile nuclease complex and may be involved in repairing other types of crosslinks that may activate the FA pathway.

Ethanol, commonly referred to as alcohol, has been a widely consumed recreational substance for centuries. However, over the years, its use has been associated with adverse health effects. Extensive research has linked alcohol consumption to over 60 diseases, including liver damage, heart disease, neurodegenerative conditions and mental illnesses[1]. Notably, alcohol consumption has been implicated in the development of several types of cancers, particularly within the digestive system. These include cancers of the oral cavity, pharynx, larynx, oesophagus, colorectal region and liver[2,3].

Alcohol metabolism involves multiple enzymes, with the χ isoform of alcohol dehydrogenase primarily responsible for converting alcohol to acetaldehyde. The liver contains the highest concentration of enzymes involved in alcohol metabolism, including acetaldehyde dehydrogenase (ALDH1 & ALDH2), which plays a crucial role in detoxifying acetaldehyde into acetate (Fig. 1A)[4]. Heavy alcohol drinkers with ALDH2 deficiency exhibited elevated levels of DNA damage, highlighting the role of ALDH2 in protecting genomic integrity[5]. Acetaldehyde possesses electrophilic properties that make it an exceptionally reactive compound, interacting with nucleophilic amino and thiol groups alike. It forms covalent bonds with

various proteins (tubulin, haemoglobin, lipoproteins, albumin and collagen), as well as with enzymes, microtubules and DNA, ultimately disrupting cell integrity[5,6].

Importantly, acetaldehyde can react with the C-2 amino group of deoxyguanosine (dG) in DNA, forming an $N^2$-ethylidene-2'-dG. This adduct can subsequently form an interstrand crosslink (ICL) with an adjacent dG[7]. The reaction with an additional acetaldehyde molecule leads to the formation of α-CH3-γ-OH-$N^2$-propano-2'-deoxyguanosine. This lesion exists in both a closed and an open form, the latter containing an aldehyde group further capable of spontaneously forming the DNA ICL, primarily with opposing guanine but also DNA-protein crosslinks[8,9] (Fig. 1B). Repairing DNA ICLs is essential for maintaining genomic integrity and preventing deleterious effects, such as cell cycle arrest, cell death or cancer. The inability to effectively repair ICLs is a hallmark of a genetic disease called Fanconi anaemia (FA)[8,10]. Seminal work has shown that acetaldehyde induces DNA damage, particularly in cells deficient in the FA repair pathway, linking acetaldehyde to DNA damage-related diseases. This research also highlights ethanol's harmful effects on foetal development, haematopoiesis, and its teratogenic potential[11].

[1]Institute of Organic Chemistry and Biochemistry of the Czech Academy of Sciences, Prague, Czechia. [2]Charles University, First Faculty of Medicine, Prague, Czechia. [3]Charles University, Faculty of Science, Prague, Czechia. ✉e-mail: silhan@uochb.cas.cz

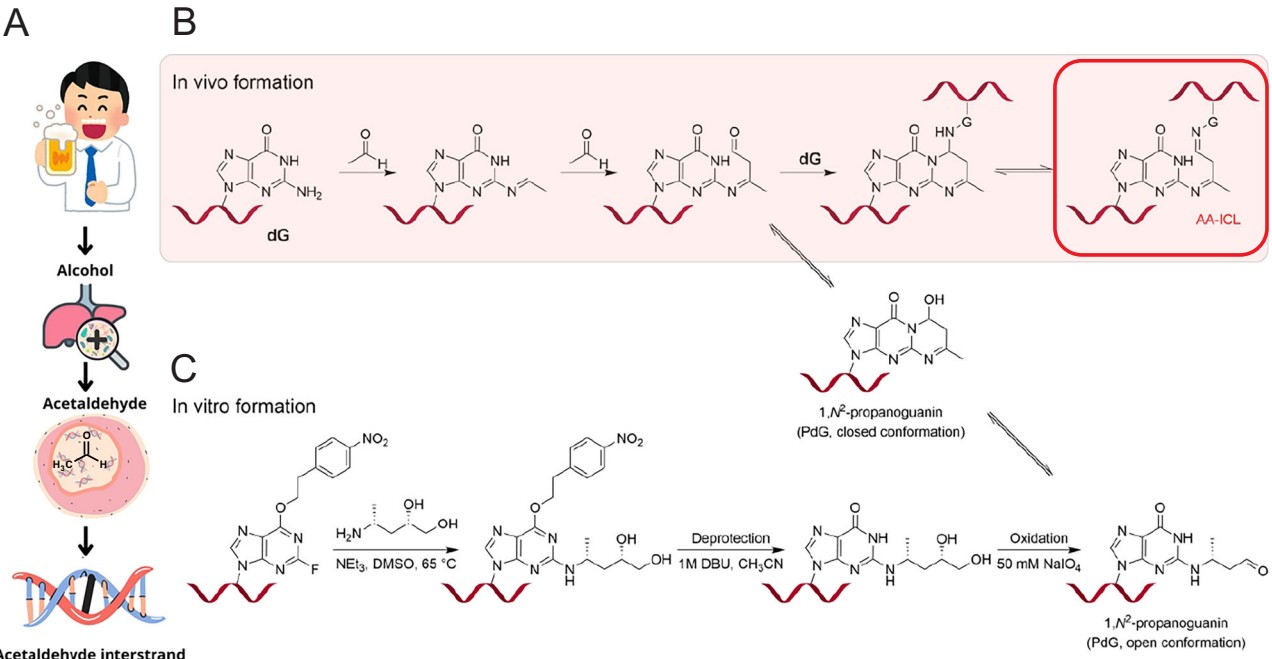

**Fig. 1 | Model of alcohol metabolism leading to acetaldehyde interstrand crosslink (AA-ICL). A** A schematic representation of the pathway from alcohol consumption through acetaldehyde to DNA damage in the form of AA-ICL (adapted from ©irasutoya, wanicon, sketchify, sketchifyedu edited via Canva.com). **B** Model of native AA-ICL formation in vivo. Two molecules of acetaldehyde react with guanosine by forming $(R)$-α-CH3-γ-OH-1,$N^2$-propano-2'-deoxyguanosine (PdG), which can subsequently form a covalent bond with another guanosine on the opposite DNA strand. **C** Synthesis of a site-specific AA-ICL using a modified oligonucleotide with 4-$(R)$-aminopentane-1,2-diol.

FA is a rare, predominantly autosomal recessive disease that results in bone marrow failure, developmental anomalies, and an increased risk of both haematologic and non-haematologic malignancies[10,12]. The hypersensitivity of cells derived from FA patients to crosslinking agents led to the discovery of the FA crosslink repair pathway, crucial for repairing ICLs during DNA replication. This discovery highlights the key role of the FA pathway in DNA repair of ICL and underscores its importance in the maintenance of genomic integrity[13-16].

When the replication machinery encounters a DNA crosslink during replication, it stalls. Due to the bidirectional nature of DNA replication, another replication machinery arrives and stalls on the opposite site. The DNA then adopts an X-shaped structure. Subsequently, ATR kinase is activated, recruiting various factors to facilitate the ICL repair[17,18]. These events activate the FA pathway, including the activation of the FA core complex, a large E3 ligase responsible for the ubiquitylation of FANCD2[19]. This monoubiquitylation is a hallmark of FA and is crucial for the subsequent excision of the ICL and the initiation of downstream repair processes[20]. Precise nuclease incisions occur when one of the DNA polymerases advances to the -1 position relative to the crosslink[17].

For ICLs induced by nitrogen mustard, the FA nuclease complex SLX4-XPF-ERCC1 (SXE) is responsible for their excision[21]. It has been shown that SLX4 stimulates the activity and specificity of the XPF-ERCC1 nuclease by approximately 100-fold towards replication-like structures. This research has demonstrated that the SXE nuclease executes two precise incisions around a synthetic nitrogen mustard ICL, identifying the nuclease responsible for the ICL incisions observed *Xenopus* egg extracts[21].

The involvement of the FA pathway in the repair of alcohol-derived crosslink suggests an explanation for acetaldehyde toxicity in the FA-deficient cell lines[11,22]. Beyond demonstrating the role of acetaldehyde in detoxification in maintaining haematopoiesis, it has been revealed that SLX4-deficient cells are hypersensitive to acetaldehyde treatment, bringing genetic evidence linking aldehyde genotoxicity with Slx4[22]. Furthermore, elevated markers for DNA damage were evident in FA-deficient cell lines, demonstrating the activation of DNA repair pathways[22]. The acetaldehyde exposure resulted in a significant reduction in the survival of SLX4-deficient, as well as other FA-deficient cells, derived from blood lineage progenitors[22]. However, the molecular details of how this lesion is removed remain unclear.

In subsequent work, we synthesised a site-specific alcohol-induced acetaldehyde ICL (AA-ICL). DNA repair experiments conducted in *Xenopus* egg extracts demonstrated that the FA pathway primarily mediates the repair of this ICL, although an alternative repair pathway was also identified[23]. Recently, a reduced form of AA-ICL was synthesised by another route, resulting in all imine bonds of the Schiff bases between the crosslink and DNA being in their reduced form. Data from this study strongly suggest that the reduced form of ICLs is exclusively repaired by the FA pathway[24].

Here, we focus on unravelling the intricate molecular details surrounding AA-ICL and its subsequent repair by the SXE complex. Specifically, we synthesised a site-specific acetaldehyde crosslink (Fig. 1C). The resulting AA-ICL contains a site-specific link between opposite DNA strands, located within the duplex of a replication fork structure, mimicking a stalled replication scenario in the presence of the crosslink. Through enzymatic assays involving SXE, we elucidate the incision mechanism employed by this nuclease complex on the AA-ICL substrate. Our results reveal the efficiency of SXE in cleaving AA-ICL. To expand our investigation, Neil3 glycosylase, a known enzyme involved in the repair of other ICLs in replication-coupled repair, is compared with SXE in the processing of the AA-ICL crosslink and another chemically distinct native ICL.

## Results

### Preparation of site-specific alcohol-induced interstrand crosslink

The synthetic crosslink in this study is chemically identical to the alcohol-induced interstrand crosslink (AA-ICL) that forms in DNA following alcohol consumption. A synthetic single-stranded DNA oligonucleotide, modified at a specific site with the 2-InoF residue, was coupled with (4 $R$)-4-aminopentan-1,2-diol via an SNAr reaction. In this reaction, the C-2 fluorine atom of the 2-InoF residue can be readily substituted by a nucleophile, such as a primary amine. The resulting product serves as the starting material for producing a

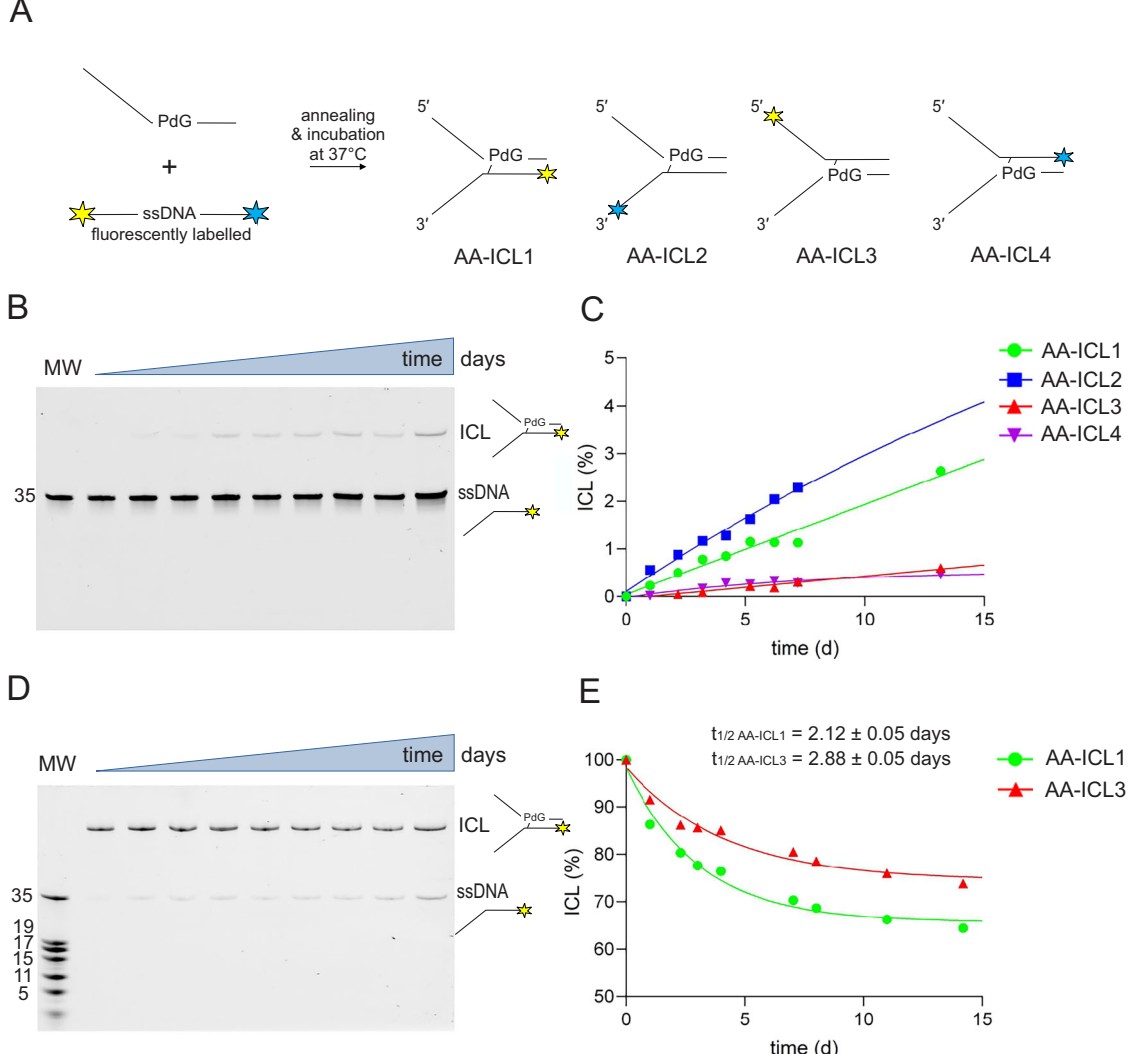

**Fig. 2 | AA-ICL formation and stability rate. A** Representation of DNA replication fork substrates used in this study differing in the position of the ICL site and its polarity. **B** 15% denaturing PAGE gel showing AA-ICL formation from annealed DNA fork. **C** The gel bands corresponding to the reactants (ssDNA) and the products of AA-ICL formation (ICL) were quantified. The relative ratios of ICL formation were plotted against reaction time for various non-crosslinked AA-ICL substrates. **D** Degradation of AA-ICL into ssDNA was monitored over time at 37 °C using a 15% denaturing PAGE gel. **E** The relative proportion of degraded AA-ICL was plotted against reaction time for various AA-ICL substrates (normalised to 100%). The graph was used to determine the half-life of ICL.

chemically identical lesion to that formed by the reaction of two acetaldehyde molecules with guanine within DNA. The 1,2-diol was converted to PdG through oxidation with $NaIO_4$. The resulting product was then purified using HPLC. The appropriate fractions were concentrated and used in subsequent crosslinking reactions (schematised in Fig. 2A).

Four different substrates generated from the original sequence were prepared to fully address the enzymatic cleavage of AA-ICL by SXE. These substrates contained an identical single-stranded DNA strand with the PdG residue prepared for crosslinking. The opposing strand was designed so the annealed oligonucleotide resembled a DNA replication fork in the shape of the letter Y. In each case, a fluorescent dye was placed at one end of the replication fork to monitor the reaction. Therefore, two 3′ and two 5′ labelled oligonucleotides were synthesised (Supplementary Fig. S1).

### Formation and stability of AA-ICL: the rate of formation of AA-ICL is relatively slow, but so is its degradation

The formation of DNA crosslinks was observed in different oligonucleotides containing the naturally identical lesion PdG (Fig. 2A). The oligonucleotides were annealed, and the crosslinking reaction was carried out in the dark. Subsamples were taken at selected intervals to determine the percentage of AA-ICL formed within the reaction mixture. The reaction was resolved on a 15% denaturing PAGE gel, and the reaction rate was 0.1% per day (Fig. 2B, C). To revalidate that the crosslink forms specifically between PdG and the opposing dG, the dG was replaced with inosine[23]. The crosslink was formed only within the duplex containing PdG with the opposing dG. In the case of the inosine-containing oligonucleotide, no crosslink was observed, confirming the necessity of the C-2 amino group (Supplementary Fig. S2).

After the crosslinking reaction, the AA-ICL was purified from the gel. To address the stability of AA-ICL, the purified crosslinked DNA was allowed to degrade over time at 37 °C. The reactions were analysed on a denaturing gel, similar to the method used for formation analysis. The half-time of two AA-ICLs was $t_{1/2\ AA\text{-}ICL1} = 2.12 \pm 0.05$ days and $t_{1/2\ AA\text{-}ICL3} = 2.88 \pm 0.05$ days. From this experiment, it was evident that degradation is again a relatively slow process, and the reaction does not reach full conversion. It appears that the crosslink forms an equilibrium with non-crosslinked DNA.

### Nuclease complex SXE unhooks alcohol-derived DNA interstrand crosslink

To elucidate the molecular mechanism by which the FA nuclease complex SXE repairs AA-ICL in enzymatic reactions, we utilised non-crosslinked

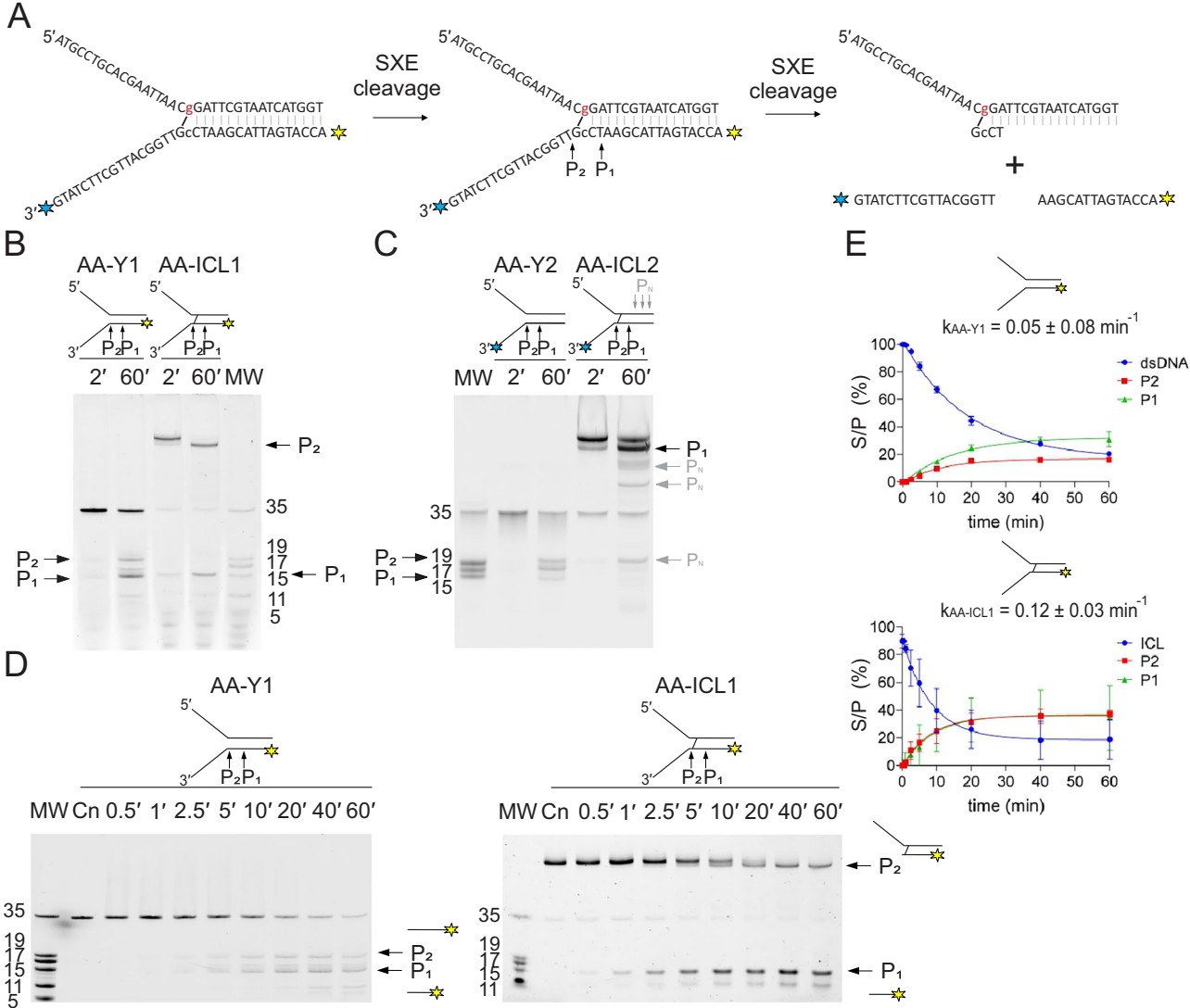

**Fig. 3 | Dual cleavage by SXE suggests unhooking of AA-ICL. A** Schematic representation of the DNA substrates and proposed SXE cleavage sites. The DNA forks were fluorescently labelled either at the 5′ end (**AA-ICL1**) or the 3′ end (**AA-ICL2**), indicated by yellow and blue labels on the bottom (leading) strand. **B, D** SXE nuclease assays (qualitative and quantitative) with non-crosslinked **Y1** and crosslinked **AA-ICL1** substrates show the formation of cleavage products $P_1$ and $P_2$. $P_1$, a short 15-nt product, appears for both crosslinked and non-crosslinked substrates.

The 19-nt product, $P_2$, from the second incision of the non-crosslinked control, migrates differently in the crosslinked **AA-ICL1** sample as it remains attached to the top strand via the crosslink, but demonstrates the second incision. **C** SXE nuclease assays (qualitative) with non-crosslinked **Y2** and crosslinked **AA-ICL2** substrates. **E** Data from **D** were plotted and fitted with exponential decay to determine reaction rates. All reactions were performed in triplicate, with error bars representing standard deviation (SD).

and crosslinked DNA forks that only differ by the presence of site-specific AA-ICL and fluorescent probes in different locations of the fork. These substrates were fluorescently labelled on either the top (lagging) or bottom (leading) strand at the 5′ or 3′ end of the left-handed fork to track cleavage events (Supplementary Fig. S1). Enzymatic reactions were resolved using 15% denaturing PAGE gel analysis. For clarity, the schematics of each crosslinked substrate and its associated SXE reaction are outlined at the top of the figure (Figs. 3A and 4A).

Fluorescent labelling of the bottom strand enabled us to visualise cleavage corresponding to this DNA strand. Qualitative reaction comparative products of cleavage of non-crosslinked and crosslinked substrates are displayed (Fig. 3B, C) with the corresponding quantitative reaction below (Fig. 3D). A persistent 35-nt band, observed in all preparations of the crosslinked substrate, likely represents a single-stranded DNA fragment resulting from partial reversal of the Schiff base linkage. Its constant intensity throughout the reaction time course indicates it is unaffected by SXE activity. This species is also evident as a product of AA-ICL

spontaneous hydrolysis (Fig. 2D). For the non-crosslinked substrate, two distinct products, 15 nt and 19 nt in size, were detected (Fig. 3B, D). We designated these incisions $P_1$ and $P_2$, confirming site-specific incisions on the bottom strand. A similar pattern of cleavage was observed for the crosslinked substrate (AA-ICL1) where one incision corresponding to $P_1$ of a size of 15 nt was observed. However, the second incision product $P_2$ migrated higher than the 35 nt control corresponding to ssDNA (Fig. 3B, D). This altered migration was consistent with one fragment being covalently crosslinked to the unlabelled arm. These results suggest that two SXE incisions occur across the DNA fork, even in the presence of a crosslink.

In contrast, the cleavage of AA-ICL2 (labelled at the 3′ end) was less efficient, with incomplete digestion of the crosslinked substrate, suggesting potential steric hindrance by the 3′ fluorophore (Fig. 3C, Supplementary Fig. S3). The products include a mirror image $P_1$ corresponding to the 15-nt arm, with AA-ICL2 migrating consistently with the larger product observed above for AA-ICL1. However, for both crosslinked and non-crosslinked AA-ICL2 substrates, the 3′ fluorescent label appears to impede the formation of the

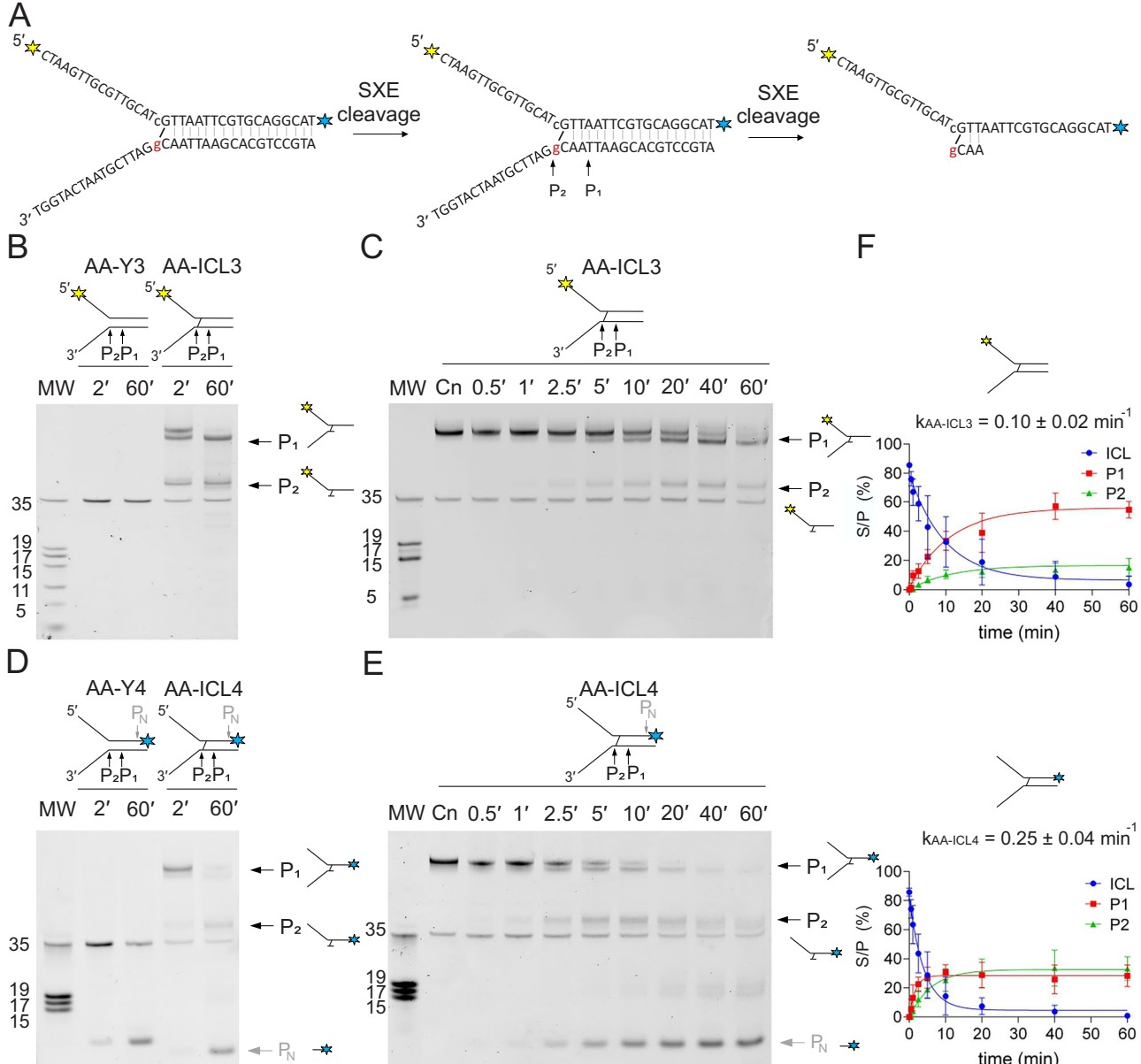

**Fig. 4 | Cleavage mechanism of AA-ICL (fluorescently labelled on the top strand) by SXE nuclease. A** Schematic of the DNA substrates and proposed SXE reaction. The fluorescent labels on the top strand are either at the 5′ end (**AA-ICL3**) or at the 3′ end (**AA-ICL4**), indicated by yellow and blue. **B** The gel depicts the reaction of SXE with non-crosslinked Y3 and crosslinked AA-ICL3. No products are visible from SXE incisions for non-crosslinked fork Y3, whilst two clear incisions $P_1$ and $P_2$ fragments are clearly visible, confirming dual incisions by SXE previously suggested in Fig. 3. Now, for AA-ICL3 previously unlabelled product of $P_2$ incision is visible. A band migrating at ~35 nt is present in all crosslinked substrate preparations, including the Cn (control) lane, and is considered a pre-existing impurity resulting from partial reversal of the Schiff base. Its relative abundance remains unaffected throughout the reaction time course, indicating it is not a product of SXE-mediated incision. **C** Time-dependent cleavage of AA-ICL3 with quantification of $P_1$ and $P_2$ products. The gel shows increased production of cleavage products with longer

incubation times. **D** Qualitative cleavage of non-crosslinked Y4 and crosslinked AA-ICL4. No products corresponding to $P_1$ and $P_2$ are visible for Y4 but dual incision is visible for AA-ICL4. For this substrate, SXE cleaves the probe at the 3′ end yielding non-specific product PN lowering the visibility of the products. **E** Time-dependent cleavage of AA-ICL4. Gel electrophoresis shows distinct bands for $P_1$ and $P_2$ over increasing reaction times, demonstrating efficient dual incision by SXE, these bands are processed by further activity of SXE removing probe at the 3′ end. It is noteworthy that visible products that are labelled $P_1$ are in fact mirror images of the cleavage and correspond to the first incision on both substrates. For all gels, the ladder is indicated to the left. **F** Kinetic analysis of SXE-mediated cleavage of AA-ICL3 (n = 5) and AA-ICL4 (n = 4). Quantitative data for $P_1$ and $P_2$ formation is plotted against time, showing cleavage efficiency. The cleavage rates ($k_{AA\text{-}ICL3}$ and $k_{AA\text{-}ICL4}$) are provided for comparison, highlighting differences in substrate processing.

second product, $P_2$, as shown in the kinetic data (Supplementary Fig. S3). Additionally, a previously unobserved minor and nonspecific product (PN) was detected (Figs. 3C, 4E). Mutational analysis of SXE revealed that these cleavage events require the enzymatically active complex (Supplementary Fig. S4). Additionally, Supplementary Fig. S4C shows that the SXE complex displays markedly increased activity compared to the XE complex when cleaving the AA-ICL3 substrate, indicating that SLX4 enhances the

endonuclease function. This is consistent with our previous findings demonstrating that SLX4 promotes the activity of the XE complex[21].

### SXE assays with AA-ICL on DNA substrate labelled on the top strand confirm dual incisions flanking the crosslink
In parallel, we tested the cleavage pattern on the fluorescently labelled top strand of the fork with labels at both ends. Reactions using a non-crosslinked

DNA fork with the 5′-end probe did not show any detectable activity of the SXE complex on the labelled strand of the substrate (Fig. 4B). However, the incised strand was not visible in the denaturing gel analysis due to the labelling setup.

In contrast, cleavage of AA-ICL3 generated two distinct products, $P_1$ and $P_2$, both migrating at molecular weights higher than the 35 nt non-crosslinked control of the entire DNA arm of the fork (Fig. 4B, C). This result indicated that a DNA crosslink (AA-ICL3) was excised on the bottom strand, as the non-crosslinked control remained unchanged by the reaction and the final product migrated at the size greater than 35 nt control. The first incision indicated as $P_1$ was the higher migrating band, where $P_2$ was likely cleaved from the other side of the crosslink and corresponds to the second incision. However, these reactions do not establish the order of the incisions.

Further investigation of the substrate labelled at the 3′ end (AA-ICL4) confirmed dual incisions by SXE. An identical pattern of bands corresponding to $P_1$ and $P_2$, as for AA-ICL3, was detected for AA-ICL4 where probes were located on opposite DNA ends, confirming the dual incision further (Fig. 4D). However, additional cleavage at the 3′ end generated a smaller product, PN, indicating nicking of the fluorophore-containing strand at the dsDNA end. Over time, as nicking continued, all bands corresponding to $P_1$ and $P_2$ disappeared (Fig. 4E), suggesting that the fluorescent label was removed by SXE from both $P_1$ and $P_2$ products. As before, the non-crosslinked DNA fork, where only the top strand was labelled, did not yield either $P_1$ or $P_2$ incisions, except for the small PN product.

## The kinetics of SXE cleavage of alcohol-derived acetaldehyde DNA interstrand crosslinks

The gels with the time-course reaction for AA-ICL were quantified and the data of cleavage were plotted against time and fitted (Figs. 3E and 4F). For labels on the top strand, specifically the 5′-labelled AA-ICL3 substrate, the cleavage rate was determined to be $k = 0.10 \pm 0.02$ min$^{-1}$, whereas the 3′-labelled AA-ICL4 exhibited a higher rate of $k = 0.25 \pm 0.04$ min$^{-1}$. This difference suggests that the position of the fluorescent label affects SXE activity, potentially due to steric hindrance by the fluorophore on the free arm of the fork. For the bottom strand, specifically the 5′-labelled AA-ICL1 substrate (Fig. 3E), the cleavage rate was determined to be $k = 0.12 \pm 0.03$ min$^{-1}$. In contrast, the cleavage rate for the non-crosslinked substrate was $k = 0.05 \pm 0.08$ min$^{-1}$. It is noteworthy that both crosslinked (ICL1) and non-crosslinked (AA-Y1) fork substrates are cleaved at comparable positions, the SXE complex displays moderately faster incision kinetics on the crosslinked substrate, with an approximately two-fold higher reaction rate (Fig. 3E).

## SXE excises the abasic site interstrand crosslinks (Ap-ICLs), a substrate of Neil3 glycosylase

SXE has been shown to cleave both AA-ICLs (this study) and nitrogen mustard ICLs[21]. To further investigate SXE's substrate range, we examined its activity on abasic site interstrand crosslinks (Ap-ICLs). Ap-ICL forms spontaneously when the aldehyde group of opened-ring ribose at an Ap site crosslinks with an Adenine (dA) adjacent to the base (dT) from the opposite strand[25] (Fig. 5A). Although analogous to AA-ICLs, Ap-ICLs exhibit fundamental differences, including their chemical composition and structural orientation. Specifically, Ap-ICLs form in the opposite direction to AA-ICLs (Supplementary Fig. S1B).

In SXE nuclease assays, we evaluated the cleavage of Y1, AA-ICL1, and Ap-ICL1 using our left-handed DNA fork substrates with the fluorescent probe at the 5′ end of the bottom strand (Fig. 5B). The control reaction with AA-ICL produced two expected products ($P_1$ and $P_2$), consistent with previous results (Fig. 3). Similarly, Ap-ICL substrates yielded in $P_1$ and $P_2$, resembling the products from AA-ICL cleavage. $P_1$ migrated significantly higher than the fork arm, while $P_2$ corresponded to a 15-nt fragment. These results confirm that Ap-ICL is unhooked and does not inhibit SXE activity. Despite differences in crosslink orientation compared to AA-ICLs, SXE cleaves Ap-ICLs efficiently. This finding demonstrates that SXE processes a broad range of DNA crosslinks, including those formed by spontaneous abasic sites.

## Mouse Neil3 enzyme responsible for Ap-ICL repair does not cleave AA-ICL

Since the Ap-ICL is repaired by Neil3 glycosylase in an upstream step before employment of the FA pathway, and SXE can process Ap-ICL consistently with other ICL, we investigated how Neil3 processes AA-ICL. We employed the catalytic NEI domain from mouse Neil3, which is responsible for the enzymatic removal of Ap-ICL during replication-coupled repair[26,27]. AA-ICL and Ap-ICL were tested to enzymatic assays of the NEI domain under the analogous reaction conditions used for enzymatic assays with SXE when comparing Ap-ICLs with AA-ICLs. The substrates (Ap-ICL2, AA-ICL1) were selected to favour Neil3 glycosylase activity, as the NEI domain exhibits a strong preference for single-stranded DNA (ssDNA) at the 3′ end of the Ap-site-containing strand[27,28].

Unlike SXE, the NEI domain did not excise or revert the AA-ICL DNA fork substrate. However, the Ap-ICL-containing fork was cleaved as expected by the glycosylase activity, serving as an internal control to validate Neil3 activity under these assay conditions (summarised in Table 1).

Overall, we have demonstrated that SXE can unhook DNA crosslinks, including those arising from alcohol consumption and acetaldehyde metabolism, a lesion known as AA-ICLs. These crosslinks are relatively stable and are efficiently excised by SXE through two distinct incisions within the DNA strand of replication fork mimetics, leaving the opposite strand intact. Importantly, we also show that SXE cleaves Ap-ICLs in a manner virtually identical to that of AA-ICLs, highlighting its ability to process a broad range of DNA crosslinks. Furthermore, our findings reinforce that SXE can also cleave other types of ICLs, such as those induced by nitrogen mustard, as previously shown[21]. These results collectively demonstrate the versatility of SXE nuclease in unhooking various DNA ICLs, presumably by recognising DNA structure rather than being selective for a specific type of crosslink. In contrast, the Neil3 glycosylase, responsible for Ap-ICL repair, does not cleave AA-ICLs, underscoring a functional distinction between the two enzymes.

## Discussion

Alcohol consumption is a major risk factor for various cancers, yet the underlying molecular mechanisms linking alcohol to genomic instability remain poorly understood. Acetaldehyde, a toxic metabolite of alcohol, reacts with DNA and can ultimately lead to the formation of interstrand crosslinks (AA-ICLs), which covalently tether the two strands of the DNA double helix. These lesions are particularly problematic because of their persistence in the genome. Their slow hydrolytic degradation leaves them intact for prolonged periods. If not repaired, AA-ICLs can block essential cellular processes like transcription and replication. This disruption leads to DNA double-strand breaks, chromosomal instability, and, ultimately, oncogenesis. Understanding how cells deal with these lesions at the molecular level is important for elucidating possible mechanisms that link alcohol consumption to genomic instability[15,22,29,30].

The number of AA-ICLs generated during alcohol consumption varies depending on factors such as alcohol intake, acetaldehyde detoxification efficiency, and genetic predispositions. Accurately quantifying these lesions remains challenging and strongly depends on the persistence of PdG (propano-deoxyguanosine), a mutagenic adduct[31–33].

Although the in vitro formation of the PdG precursor is relatively inefficient, constituting less than 10% of the primary acetaldehyde adduct N2-ethylidene-dG, excessive acetaldehyde exposure increases the PdG formation rate to approximately 1 PdG per $1.3 \times 10^8$ dG per 24 h[34]. In a diploid human genome ($1.23 \times 10^9$ GC pairs), this results in about 20 PdG molecules formed per day. Moreover, the presence of basic molecules such as histones and polyamines, including spermidine, significantly enhances PdG formation, indicating a strong environmental influence[35].

Acetaldehyde typically binds to intracellular macromolecules by forming imines, or Schiff bases, these adducts are in dynamic equilibrium with their unreacted components and are prone to hydrolysis under physiological conditions[36,37]. In this study, we demonstrated the formation and decomposition rates of the AA-ICL. The isolated, relatively pure AA-ICL

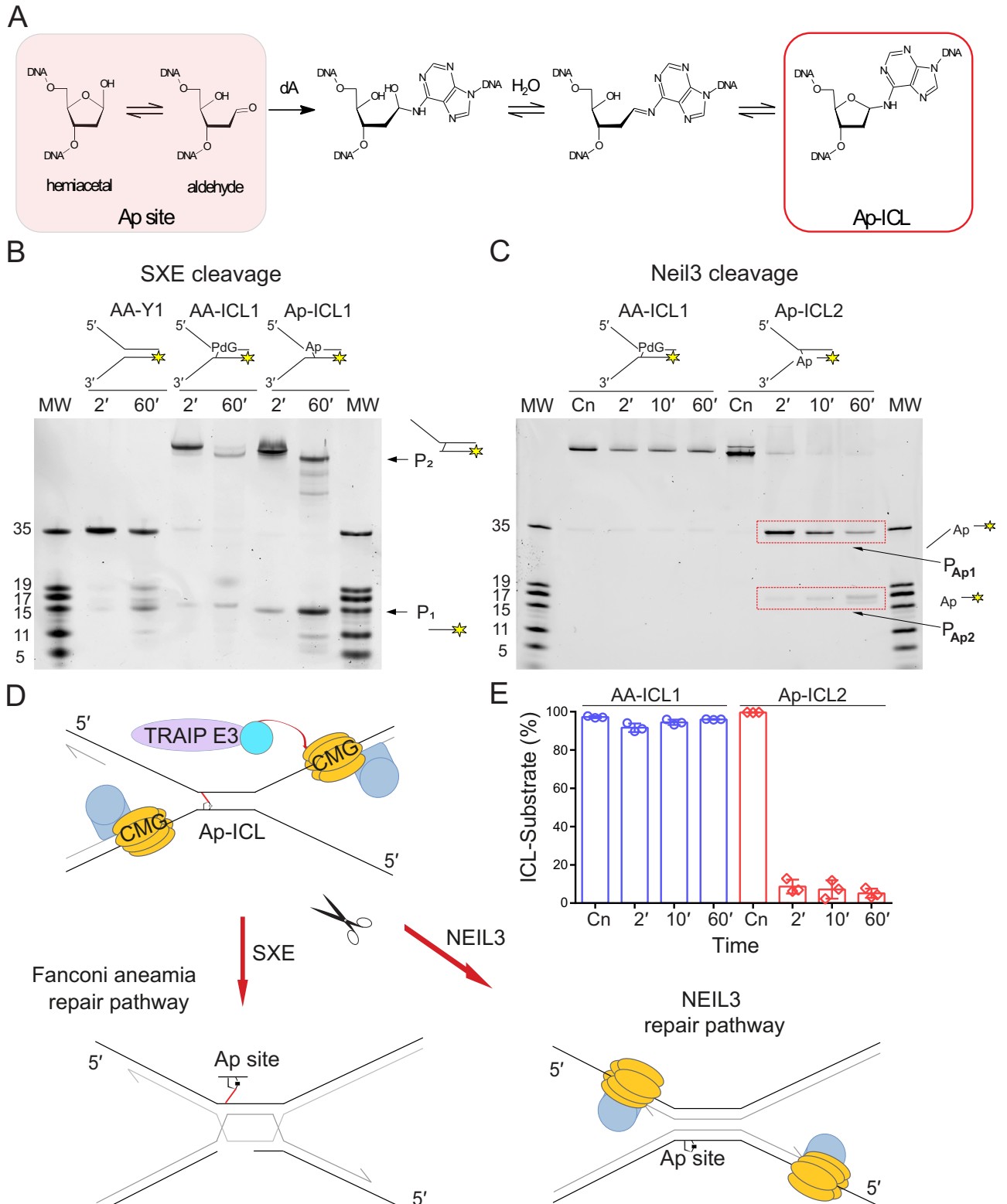

**Fig. 5 | Comparison of cleavage mechanism of AA-ICL and Ap-ICL. A** The schematics of the Ap-ICL formation. **B** SXE nuclease assay comparing Y1, AA-ICL (AA-ICL1), and Ap-ICL (Ap-ICL1) with respective nuclease products $P_1$ and $P_2$. **C** Neil3 glycosylase (NEI domain) assay with the AA-ICL (AA-ICL1) and Ap-ICL (Ap-ICL2). The alcohol-derived crosslink AA-ICL remains unprocessed by the enzyme unlike Ap-ICL, which is quickly turned into products. **D** A schematic model of Ap-ICL cleavage and potential repair mechanisms. Both Neil3 and the FA repair pathway recognise and process Ap-ICL. In the FA pathway, Ap-ICL is cleaved similarly to AA-ICL by the SXE nuclease, resulting in double-strand breaks that serve as substrates for further repair steps, such as homologous recombination. In contrast, during the Neil3 repair pathway, the helicase is not unloaded from the DNA, and Neil3 unhooks the Ap-ICL. **E** Quantification of the Neil3 cleavage assays where the depletion of substrates from panel C was plotted against time. Reactions were repeated in triplicate, and the error is represented as standard deviation (SD). The ICL incision data are qualitatively summarised in Table 1.

## Table 1 | Relative incision efficiencies of SXE versus NEIL3

| Substrate/Enzyme | SXE | NEIL3 |
|---|---|---|
| AA-ICL | ✓ High incision (++) | ✗ No incision (−) |
| Ap-ICL | ✓ High incision (++) | ✓ Very high incision (+++) |

Relative qualitative incision efficiency, based on gel band intensities and known enzymatic activity. Neil3 only cleaves Ap-ICLs when the abasic (Ap) site is adjacent to a single-stranded DNA region on its 3′-side.

decomposed slowly and did not reach equilibrium even after 14 days, indicating that the AA-ICL is remarkably stable, with approximately 70% of the original AA-ICL still remaining in the reaction mixture. Although the isolated AA-ICL decomposed slightly during purification, as indicated by a small percentage of ssDNA, the level of ssDNA remains consistent throughout the reactions. These stable background signals could have been subtracted during analysis to focus on enzymatic activity and product formation. Time-course experiments confirmed that the band corresponding to the non-crosslinked strand remains stable, enabling accurate subtraction and ensuring a clear focus on enzymatic processes.

As previously demonstrated for nitrogen mustard ICLs, the SXE complex cleaves ICLs through a dual incision, making two precise cuts on one strand of the replication fork around the crosslink, not dissimilar to activity observed for XE alone[21,38]. It was shown that Slx4 significantly enhances the activity of the SXE complex, particularly towards forked DNA structures, thereby altering substrate specificity[21]. Notably, SXE exhibits no significant preference for crosslinked over non-crosslinked substrates.

While the FA pathway and an unknown mechanism have been implicated in AA-ICL repair in *Xenopus* egg extracts, the precise molecular mechanisms and enzymes responsible remain unclear[23]. Here, we demonstrate that SXE unhooks AA-ICLs by making two incisions around the crosslink, specifically on the bottom strand of the replication fork containing 3′-ssDNA arms. This confirms that in vitro SXE can cleave AA-ICLs in a manner similar to nitrogen mustard DNA ICLs.

Interestingly, the reaction rates of SXE-mediated incisions remain consistent regardless of the presence of a crosslink, suggesting that SXE primarily recognises forked DNA structures rather than the ICL itself. We speculate, that under in vivo conditions, additional factors, such as FA proteins and the X-structure of the replication fork, may influence SXE's specificity and efficiency. The incisions observed in our study, in the context of the X-structure, would lead to the unhooking of the AA-ICL and the separation of DNA strands. This separation would allow the damaged strands to undergo subsequent repair steps via the FA pathway, a process critical for restoring replication, maintaining genomic stability, and ensuring proper cellular function.

In our previous work, we have shown that AA-ICL repair occurs via two distinct mechanisms: the FA pathway and an as-yet-unknown mechanism[23]. Here, we provide complementary data to the work in *Xenopus* egg extracts and further demonstrate that, unlike Ap-ICLs, Neil3 does not cleave AA-ICLs. This finding further supports the idea that the unknown mechanism of AA-ICL repair does not involve the Neil3 pathway.

Here, we also show that SXE cleaves the Ap-ICL and, therefore, is not specific to crosslinks but rather it cleaves around the damaged site, regardless of the nature or presence of the ICL[21]. Moreover, a recent study has shown that FA cells are highly sensitive to glutathione (GSH) depletion, as GSH detoxifies small reactive aldehydes like formaldehyde and helps mitigate oxidative stress[39]. While the direct link between GSH depletion and Ap-ICL or aldehyde-induced ICL formation remains unclear, earlier research suggests a connection between GSH depletion and increased ICL formation[40]. The oxidative damage under low GSH conditions may lead to more ICLs, which are then repaired by the SXE complex. This correlation may help explain the heightened sensitivity and lethality observed in FA-deficient cells under the GSH depletion. However, the specific contribution

of SXE to ICL repair in vivo under these conditions requires further investigation.

Neil3 is the primary enzyme responsible for repairing Ap-ICLs and similar lesions[26,27]. Should this mechanism fail, the FA may be employed for the repair pathway of the ICL[41–43]. Early studies on psoralen-ICL formed under UV radiation showed that XE nuclease cleaves this crosslink[38], engaging in a similar unhooking event as for Ap-ICL[26,27]. It is conceivable that SXE could also process psoralen-ICLs in a manner similar to other ICLs, making two sequential incisions on the bottom strand while leaving the other DNA strand intact.

Our in vitro findings demonstrate that SXE is capable of cleaving alcohol-derived AA-ICLs and Ap-ICLs in a comparable manner, suggesting a potential role as a versatile FA nuclease. However, further studies are needed to confirm its involvement in AA-ICL repair in cells. SXE can cleave a broad range of ICLs, including those induced by acetaldehyde, abasic sites, nitrogen mustard, and possibly others, establishing it as a powerful tool for ICL unhooking. Consequently, these findings raise the possibility that the FA repair pathway emerges as a universal mechanism capable of addressing various types of ICLs.

## Methods

### DNA oligonucleotides preparation

All DNA oligonucleotides were synthesised commercially (Eurofins Genomics). The sequences of all DNA oligonucleotides and structures of annealed substrates are provided in the supplementary file (Supplementary Fig. S2). Fluorescently labelled oligonucleotides were purified by band excision from 15% PAGE gel, followed by elution into TE buffer (10 mM Tris pH 8.0, 1 mM EDTA). Purified samples of ssDNA were subsequently concentrated on Amicon 0.5 with molecular cutoff 3 kDa (Cytiva Micro-SpinTM G-25) to approx. 10 uM.

### Synthesis of aminopentadiol

The synthesis of (4R)-4-aminopentane-1,2-diol has been described previously[23]. Briefly: First, the commercially available (S)-pent-4-en-2-ol was treated with phthalimide under the Mitsunobu conditions to afford the protected (R)-pent-4-en-2-amine in 75–85% yield. The phthalimide protection was exchanged for a benzyloxycarbonyl (Cbz) in a one-pot, three-step sequence affording the Cbz-protected intermediate in 56–63% yield. The terminal double bond was dihydroxylated using the $RuCl_3$-$CeCl_3$ catalytic system and sodium periodate as an oxidising agent. The resulting diol (49% yield) was isolated as a diastereomeric mixture (ca 1:1). Cleavage of the Cbz protecting group was achieved hydrogenolytically on a Pearlman's catalyst. No chromatographic purification of the free amine was necessary in this final step. The overall yield of the synthesis was slightly lower than reported in the literature. Achieved purity was >95% (NMR-based). The scheme of the synthesis is provided in the supplementary file (Supplementary Fig. S2).

### Preparation of AA-ICL on fluorescently labelled substrate

The following procedure was performed according to our previous work[23]. Briefly, a 35-nt oligonucleotide ATGCCTGCACGAATTAAC[2-F-dI-CE]GATTCGTAATCATGGT (2-F-dI donates 2'-Deoxy-2'-fluoroinosine, supplied by Eurogentec SA) immobilised on solid support was incubated with (4R)-4-aminopentane-1,2-diol prepared as described above (Fig. 1C).

The O-6 protecting group was removed by reaction with DBU. The remaining protecting groups of the oligonucleotide were removed by a 28% aqueous solution of ammonia, and the oligonucleotide was eluted from the solid support. The aqueous solution was rapidly frozen in liquid nitrogen and freeze-dried using a benchtop lyophilization system FreeZone Plus 2.5 Plus (Labconco Corporation). The sample was subsequently dissolved in 0.1 M TEAAc, purified by HPLC on a semi-preparative column Phenomenex Luna 5 μm C18 column (150 mm × 10 mm) and re-lyophilised. The vicinal diol was oxidatively cleaved with 50 mM sodium periodate. The aqueous solution was rapidly frozen in liquid nitrogen, freeze-dried, and purified on HPLC to afford the final oligonucleotide. The identity of the product was validated by MS (Supplementary Fig. S2).

The modified DNA oligonucleotide (5'-ATGCCTGCACGAAT-TAACG*GATTCGTAATCATGGT3'), containing the $(R)$-α-CH3-γ-OH-1,$N^2$-propano-2'-deoxyguanosine (G*), was mixed with a partially complementary, fluorescently labelled DNA oligonucleotide (Fig. 2A, sequence details in Supplementary Fig. S2). The mixture was annealed by slow cooling from 95 °C in the substrate buffer (20 mM HEPES pH 7.0, 150 mM NaCl). The reaction mixture was incubated at 37 °C for several days, followed by the isolation of DNA duplex with the ICL from a 15% denaturing PAGE gel.

### Abasic interstrand crosslink formation and isolation

Abasic interstrand crosslink (Ap-ICL) within the DNA duplex was prepared and isolated as described previously[44]. Briefly, labelled and unlabelled complementary oligonucleotides were combined in equimolar ratios in a buffer (20 mM HEPES, pH 6.5, 140 mM NaCl, 0.5 mM TCEP, and 5% glycerol). The reaction mixture was annealed by heating to 95 °C followed by gradual cooling to ambient temperature. To generate the Ap site, 0.5 units of uracil-DNA glycosylase (New England Biolabs) was introduced and incubated at room temperature for 5 min. The resulting DNA fork containing the Ap site was incubated at 37 °C to crosslink formation. The Ap-ICL was isolated from the polyacrylamide gel using a modified electrophoretic band excision protocol[45].

### Cloning, expression and purification of recombinant Slx4-Xpf-Ercc1

The mouse cDNA for *Slx4*(1-758) and *Ercc1*(HT-3C-Full length) were cloned to pAcebac1 vectors, and the mouse cDNA for *Xpf* (full length) was cloned to pIDC vector. The Xpf and Ercc1 constructs for expression of Xpf and Ercc1 proteins were fused by Cre recombinase (New England Biolabs). The constructs for Slx4 and Xpf-Ercc1 were then transformed to *E. coli* DH10EMBacY (Genova Biotech), and isolated bacmids were transfected to Sf9 cells using Fugene6 and 24-well plates with $1 \times 10^6$ cells. Secondary recombinant baculovirus was used to *co*-infect Sf9 insect cells (2 L) at a density of $2–3 \times 10^6$ cells/ml, and the culture was grown for 68 h before being harvested. All purification steps were carried out in a buffer containing 20 mM Tris pH 8.0, 150 mM–1 M NaCl, 10% (*v/v*) glycerol, and 3 mM B-ME. Cells were homogenised by sonication, followed by affinity chromatography using an MBP-tagged domain. Proteins were eluted with the buffer supplemented with 20 mM maltose. The complexes were then loaded onto a HiTrap™ Heparin HP 5 ml column (GE Healthcare) and eluted using a NaCl gradient. The MBP tag was cleaved overnight at 4 °C using TEV protease, while the HT tag was cleaved using 3C protease. Concentrated samples were further purified on a Superose 6 Increase 10/300 GL column (Cytiva), and the combined fractions were flash-frozen. Xpf point mutants (primers for mutation are provided in Supplementary Fig. S4) in the XE and SXE complexes were purified using a procedure identical to the wild type.

### Cloning, expression, and purification of Neil3

Cloning, expression, and purification processes were described previously[27]. Briefly, the NEI domain was cloned into a modified pET-24b vector that includes the C-terminal 3 C protease (HRV) site and subsequent His6x tag. The plasmid was transformed into Escherichia coli BL21 StarTM (DE3) cells (ThermoFisher). An initial 5 ml culture was grown overnight in LB medium at 37 °C. Protein expression was conducted in ZY5052

autoinduction media supplemented with 50 μM ZnSO4. The culture was grown at 37 °C until $OD_{600} = 0.6–1$; then, the temperature was lowered to 18 °C for overnight growth.

Harvested cells were disrupted via sonication in a lysis buffer comprising 20 mM Tris-HCl (pH 8.0), 300 mM NaCl, 30 mM imidazole pH 8.0, 10% (*v/v*) glycerol and 1 mM TCEP. The lysate was fractionated by centrifugation, and the supernatant was incubated with Ni-NTA resin (Machery-Nagel) using the batch technique. Protein was eluted using an imidazole-enriched buffer.

Protein was desalted using a HiPrep 26/10 desalting column, followed by fractionation via cation exchange chromatography on a HiTrap SP HP column. The buffer A mobile phase consisted of 20 mM Tris-Hcl pH 8.0, 70 mM NaCl, 10% (*v/v*) glycerol and 2 mM B-ME. Protein was eluted using a NaCl gradient. Final purification was achieved via size-exclusion chromatography using a Superdex 75 Increase GL HiLoad 10/300 column equilibrated with buffer A.

Protein purity was verified by SDS-PAGE on a 15% polyacrylamide gel. The purified NEI domain was concentrated, flash-frozen in liquid nitrogen and stored at −80 °C.

### Kinetics of AA-ICL formation and stability

The reaction mixture was incubated at 37 °C under near-physiological conditions (20 mM HEPES, pH 7.0; 150 mM NaCl) for several days. Periodic aliquots were taken for stability assays. Afterwards, DNA duplexes containing ICLs were purified by isolation from 15% denaturing PAGE gel. The isolated ICLs were diluted in substrate buffer and subjected to a stability assay at 37 °C the aliquots were taken at given timepoints. Subsequently, the aliquots were analysed using a 15% denaturing gel.

### Nuclease assay

All reactions were carried out in a nuclease buffer (25 mM Tris pH 8.0, 50 mM NaCl, 2 mM MgCl2, 1 mM TCEP, 0.1 mg/ml BSA, 5% glycerol) at 25 °C. The crosslinked substrate was synthesised as described above. Reactions were initiated by mixing 40 nM of the given substrate with 100 nM of the enzyme SXE. After the incubation period, the reactions were quenched with 80% formamide, 200 mM NaCl, 10 mM EDTA, 0.01% bromophenol blue and analysed on 15% denaturing PAGE gel. The gels were visualised using the Amersham Typhoon laser scanner (Cytiva). The signals were analysed using ImageQuant TL version 8.2.0 software (Cytiva). The reaction kinetics were plotted in GraphPad Prism version 8.0.1 and fitted using a one-phase decay model, and the complete fitting statistics are shown in Supplementary Data.

### Glycosylase assay

Enzymatic assays with the NEI domain and DNA substrates containing Ap-ICL or AA-ICL were performed in parallel with nuclease assays in Neil3 reaction buffer (20 mM Tris-HCl, pH 7.4, 75 mM NaCl, 5% glycerol, 0.1 mM TCEP). Enzyme and substrate were used at final concentrations of 20 nM and 40 nM, respectively. Reactions were initiated by mixing equal volumes of enzyme and substrate at 25 °C. At designated time points, aliquots were withdrawn and quenched with an equal volume of quenching buffer (80% formamide, 40 mM EDTA). Samples were resolved by 20% denaturing PAGE (20% acrylamide:bisacrylamide 19:1, 7 M urea, 1× TBE). The gels were evaluated identically to those with nuclease assay.

### Statistics and reproducibility

All quantitative data were analysed using standard statistical methods as indicated in figure legends. Where applicable, means ± standard deviation (SD) are shown, while individual datapoints are provided in Supplementary Fig. S5. Experiments were repeated independently at least three times unless otherwise stated, and all replicates refer to biological replicates, independently prepared and processed samples, not repeated measurements of the same sample. Gel-based assays were performed independently, including with freshly prepared substrate. The representative gels are shown, and uncropped data are shown in Supplementary Fig. S6. Band intensities were

quantified using ImageQuant TL, and results were consistent across all biological replicates. Sample sizes and statistical approaches used for data presentation and comparison are described in the figure legends.

## Reporting summary
Further information on research design is available in the Nature Portfolio Reporting Summary linked to this article.

## Data availability
All data analyses are presented in the main text or supplementary materials and supplementary data. All source data underlying the graphs and charts presented in the main figures are presented as Supplementary Data (either as standalone graphs, excel tables or in text format).

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

## Acknowledgements
The work was supported by the Czech Science Foundation [24-12306S]; Institute of Organic Chemistry and Biochemistry; Academy of Sciences Czech Republic [RVO: 61388963]. Funding for open access charge: Academy of Sciences Czech Republic [24-12306S].

## Author contributions
J.H. and A.H. conducted experiments, analysed the data, and designed the figures. M.D. performed chemical synthesis with the aid from J.H. and A.A. E.B. and R.N. supervised their work. J.S., J.H. and A.H. wrote the manuscript. J.S. conceptualised and supervised the study and secured funding.

## Competing interests
The authors declare no competing interests.
