## [Transparent Peer Review file · Communications Biology]

Mechanistic Insights into Alcohol-Induced DNA Crosslink Repair by Slx4-Xpf-Ercc1 Nuclease Complex in the Fanconi Anaemia Pathway

Corresponding Author: Dr Jan Silhan

Version 0:

Reviewer comments:

Reviewer #1

(Remarks to the Author)

Silhan and collaborators address the contribution of the Slx4-Xpf-Ercc1 (SXE) complex and the Neil3 DNA glycosylase to the repair of 2'-propanodG- or AP-site- interstrand crosslinks (PdG- or AP- ICLs). The authors chemically synthesized oligos containing site specific PdG-ICL in a replication fork-like DNA structure and address the contribution of these DNA processing enzymes on AA-ICL or AP-ICL unhooking. They found that SXE complex incise twice on the PdG-containing DNA strand, thus recapitulating the key nucleolytic steps of the FA pathway during acetaldehyde genotoxicity. Similarly, they also found the SXE complex incises both AA-ICLs and AP-ICLs. Moreover, the NEIL3 catalytic domain does is efficient at cleaving AP-ICL while it does not show any activity on AA-ICLs. Overall, these studies are scientifically sounded. The quality of the experimental approaches, the results obtained, and their interpretation are potentially interesting for the DNA repair and genome instability fields. However, there are a few concerns about the work, which need further clarifications before publication:

1-. Related to Figure 3C and Figure 3E. The authors describe that the SXE complex makes two incisions P1 and P2 in both non-crosslinked AA-Y1 and AA-crosslinked ICL1 substrates. However, the authors do not describe whether the SXE complex processes crosslinked and non-crosslinked replication fork substrates at the same rates or SXE is more efficient at crosslinked substrate. To this reviewer, it seems that SXE complex is catalytically more active on crosslinked substrates. Could the authors state this from their reaction rate curves?

2-. Related to Figure 4B: The authors claim in figure 4B that the appearance of a 35-nt long product observe in lanes 1 and 2 are unprocessed upper labeled DNA oligo as incision are to happen in the bottom strand. However, in lanes 3 and 4 there is a 35-nt band despite of the P2 product formed. Could the authors speculate what the nature of this 35-nt band is? Please include an explanation for this in the text.

3-. Related to figure 4D. The authors calculated KAA-ICL4. Could the authors clarify how this was calculated? Do they take into consideration the SXE activity on the fluorophore DNA sequence for their calculations?

4-. Pg 14. Ln 336. The authors obtained a Ap-ICL by reacting dU-excised AP-site with dA in the opposite strand. However, Figure S2B table shows that YpYA1 or ApYB2 contains a dT in form of the AP-site. Please clarify this issue.

Minor points:

Pg. 1, In 16: use "while" instead "and".

Pg. 2, In 38: "alcoholics" should be substituted by "heavy drinkers" or similar expression.

Pg. 5, In 124: "35 NK". I guess the authors meant 35-nt.

Pg. 5, In 130: "dissolved in 1 dM" should be written as "dissolved in 0,1 M" or dissolved in 100 mM for simplicity.

Pg. 5, lines 151 and 152: "The mouse genes" should be corrected to "The mouse cDNAs"

Reviewer #2

(Remarks to the Author)

Employing a synthetic DNA substrate containing a site-specific acetaldehyde interstrand crosslink (AA-ICL) located within a stalled replication mimic, the authors performed enzymatic assays with SLX4-XPF-ERCC1 (SXE) to elucidate the incision mechanism employed by this nuclease complex. First, the formation and stability of the AA-ICL substrate was investigated, revealing both slow generation and slow degradation. Second, the molecular products and efficiency of the SXE nuclease was determined on non-crosslinked and crosslinked synthetic AA-ICL and abasic site (Ap)-ICL substrates with varying positioned fluorescent labels. Third, comparison analysis indicated a wide range of DNA substrates for the SXE nuclease complex, whereas the Neil3 glycosylase, responsible for Ap-ICL repair, does not cleave AA-ICLs, underscoring a functional distinction between the two repair mechanisms. Overall, the study represents a straightforward and generally well executed collection of work, offering fundamental insights into biochemical repair of ICLs. For consideration:

The authors appear to suggest that the reactions of ICL substrate formation or degradation do not go to completion (Fig. 2). Has this been attempted, i.e., have the reactions been carried out for months, instead of days? Do the authors have an explanation for why the reactions may not go to completion (is it possible a percentage of the substrate cannot react for some reason)? That could be explored further in the Discussion.

It's unclear whether the authors have compared SXE versus XE incision patterns and efficiency on the full range of substrates. Since prior work has argued for a stimulatory role for SLX4, analysis of this sort might prove insightful.

Inclusion of a Table that qualitatively summarizes the relative incision efficiencies of SXE versus NEIL3 would be a nice and easy to follow reference.

Reviewer #3

(Remarks to the Author)

Havlikova et al investigates the enzymatic activity of the Slx4-Xpf-Ercc1 (SXE) nuclease complex on synthetic acetaldehyde-induced DNA interstrand crosslinks (AA-ICLs). The authors synthesise site-specific AA-ICLs and demonstrate that SXE can cleave these substrates through dual incisions flanking the crosslink. They also show that SXE cleaves abasic site interstrand crosslinks (Ap-ICLs) but that Neil3 glycosylase does not process AA-ICLs. The experimental work is technically competent, and some very nice biochemical results are shown, but could do with a little toning down of the biological significance.

The main findings of the paper are well supported, that a purified SXE complex can cleave adjacent to an AA-ICL, but that interestingly NEIL3 glycosylase cannot. This demonstrates that like some other ICL agents (eg platinum and nitrogen mustards) only the FA pathway can unhook the damage. The figures as presented are excellent, and well controlled, and the experimental tools developed will be useful to the field of DNA repair research.

However, the contribution of SLX4 is only referenced via previous work – and SLX1 is omitted entirely from the experiments (likely because the SLX4 utilised is a truncated form that doesn't bind SLX1). It is highly likely that XPF:ERCC1 would perform exactly the same cleavage as the SXE proteins used here, based on previous publications showing it can cleave branched DNA with same cleavage profile as presented (on other structures, but not AA-ICL). Do the authors have results comparing XE and SXE activity in at least one experiment; showing the contribution of SLX4 (if any) would improve the significance of the work.

Moreover, the authors have not provided evidence that SXE is actually responsible for repairing these crosslinks in cells - only that it can cleave them in vitro. As such, it is not responsible to make major conclusions about the FA pathway in response to ICLs generated by "alcohol" when no data exists that directly link the pathway to AA-ICL repair (as opposed to some protein:DNA or other DNA adduct). The introduction and discussion should be toned down and a clearer distinction should be drawn between what has been demonstrated experimentally versus what is speculated. The discussion overstates the implications of the findings for understanding alcohol-related disease.

Otherwise, a nice strong biochemical paper!

Version 1:

Reviewer comments:

Reviewer #1

(Remarks to the Author)

My scientific concerns have been properly addressed by the authors. I have no further comments.

Reviewer #2

(Remarks to the Author)

The reviewers have sufficiently addressed my previous concerns.

Reviewer #3

(Remarks to the Author)

The authors have addressed my concerns.

Point-to-point responses to reviewers' comments

Mechanistic Insights into Alcohol-Induced DNA Crosslink Repair by Slx4-Xpf-Ercc1 Nuclease Complex in the Fanconi Anaemia Pathway

Jana Havlikova^{1,2}, Milan Dejmek¹, Andrea Huskova¹, Anthony Allan³, Evzen Boura¹, Radim Nencka¹, Jan Silhan^{1,*}

Reviewers' comments:

Reviewer #1 (Remarks to the Author):

Silhan and collaborators address the contribution of the Slx4-Xpf-Ercc1 (SXE) complex and the Neil3 DNA glycosylase to the repair of 2'-propanodG- or AP-site-interstrand crosslinks (PdG- or AP- ICLs). The authors chemically synthesized oligos containing site specific PdG-ICL in a replication fork-like DNA structure and address the contribution of these DNA processing enzymes on AA-ICL or AP-ICL unhooking. They found that SXE complex incise twice on the PdG-containing DNA strand, thus recapitulating the key nucleolytic steps of the FA pathway during acetaldehyde genotoxicity. Similarly, they also found the SXE complex incises both AA-ICLs and AP-ICLs. Moreover, the NEIL3 catalytic domain does is efficient at cleaving AP-ICL while it does not show any activity on AA-ICLs. Overall, these studies are scientifically sounded. The quality of the experimental approaches, the results obtained, and their interpretation are potentially interesting for the DNA repair and genome instability fields. However, there are a few concerns about the work, which need further clarifications before publication:

1-. Related to Figure 3C and Figure 3E. The authors describe that the SXE complex makes two incisions P1 and P2 in both non-crosslinked AA-Y1 and AA-crosslinked ICL1 substrates. However, the authors do not describe whether the SXE complex processes crosslinked and non-crosslinked replication fork substrates at the same rates or SXE is more efficient at crosslinked substrate. To this reviewer, it seems that SXE complex is catalytically more active on crosslinked substrates. Could the authors state this from their reaction rate curves?

We thank the reviewer for the careful reading of our manuscript and for this valuable observation. Our data indeed indicate a modest preference of the SXE complex for the crosslinked ICL1 substrate compared to the non-crosslinked AA-Y1 substrate. As the difference in reaction rates was approximately two-fold, we initially did not consider it substantial enough to highlight. However, in light of the reviewer's helpful suggestion, we have re-evaluated our interpretation and revised the manuscript to acknowledge this catalytic preference. The corresponding section of the main text have been updated to reflect this point.

Updated sentence for the manuscript (page 12, line 29-31):

“It is noteworthy that both crosslinked (ICL1) and non-crosslinked (AA-Y1) fork substrates are cleaved at comparable positions, the SXE complex displays moderately faster incision kinetics on the crosslinked substrate, with an approximately two-fold higher reaction rate (Figure 3E).”

2-. Related to Figure 4B: The authors claim in figure 4B that the appearance of a 35-nt long product observe in lanes 1 and 2 are unprocessed upper labeled DNA oligo as incision are to happen in the bottom strand. However, in lanes 3 and 4 there is a 35-nt band despite of the P2 product formed. Could the authors speculate what the nature of this 35-nt band is? Please include an explanation for this in the text.

We thank the reviewer for this insightful comment. The 35-nt band observed in lanes 3 and 4 is likely due to the reversible nature of the Schiff base crosslink, which can result in partial hydrolysis or incomplete assembly of the DNA substrate. This leads to the release of a single-stranded DNA arm corresponding to one side of the fork, migrating at 35 nucleotides. Importantly, this band is not a result of enzymatic incision, as it is also present in the control (CN) lane. The appearance of this band is consistent with our observations in Figure 2D, where it is annotated as ssDNA. We have now clarified this point in the main text and figure legends.

Updated sentence for the manuscript (page 10, line 9-13):

“A persistent 35-nt band, observed in all preparations of the crosslinked substrate, likely represents a single-stranded DNA fragment resulting from partial reversal of the Schiff base linkage. Its constant intensity throughout the reaction time course indicates it is unaffected by SXE activity. This species is also evident as a product of AA-ICL spontaneous hydrolysis (Figure 2D).”

In Figure 4 legend (page 13, line 8-11) we state:

“A band migrating at ~35 nt is present in all crosslinked substrate preparations, including the CN (control) lane, and is considered a pre-existing impurity resulting from partial reversal of the Schiff base. Its relative abundance remains unaffected throughout the reaction time course, indicating it is not a product of SXE-mediated incision.”

3-. Related to figure 4D. The authors calculated KAA-ICL4. Could the authors clarify how this was calculated? Do they take into consideration the SXE activity on the fluorophore DNA sequence for their calculations?

We apologise for the lack of information. We now have updated materials and methods stating the fitting of the data:

Specifically, now we state in the manuscript (page 6, line 32-34): *“The reaction kinetics were plotted in GraphPad Prism version 8.0.1 and fitted using a one-phase decay model, and the complete fitting statistics are shown in Supplementary Data.”*

SXE activity toward the fluorophore-labeled DNA sequence was not considered in the calculations, since cleavage was observed in all products.”

4-. Pg 14. Ln 336. The authors obtained a Ap-ICL by reacting dU-excised AP-site with dA in the opposite strand. However, Figure S2B table shows that YpYA1 or ApYB2 contains a dT in form of the AP-site. Please clarify this issue.

Thank you for highlighting this point. We appreciate your careful reading and recognise that the original description of the AP-ICL formation was ambiguous.

As shown in Supplementary Figure S2B, the oligonucleotide sequence contains a thymine (dT) opposite the abasic (AP) site, but the crosslink does not form directly with the thymine. Instead, the AP site is generated by uracil excision (via UDG treatment) on one strand, and the crosslink forms covalently between this abasic site and an adenine residue located on the opposite strand, specifically the adenine adjacent (5' upstream) to the thymine.

To clarify, the oligonucleotide pairs ApYA1 + ApYB1 (forming duplex YAP1) and ApYA2 + ApYB2 (forming duplex YAP2), as detailed in Supplementary Figures S2B and S3, are designed such that UDG treatment creates the abasic site. This site then reacts with the neighbouring adenine on the opposite strand to form the AP-ICL, as illustrated in Supplementary Figure S3.

We have updated the Figure S3 legend as follows:

“The AP-ICL is formed by a covalent linkage between an abasic (AP) site generated by uracil excision by UDG and an adenine residue on the opposite strand. This adenine is located adjacent to an orphaned thymine (dT) opposite the AP site, specifically positioned 5' to the thymine.”

Minor points:

Pg. 1, Ln 16: use “while” instead “and”.

Pg. 2, Ln 38: “alcoholics” should be substituted by “heavy alcohol drinkers” or similar expression.

Pg. 5, Ln 124: “35 NK”. I guess the authors meant 35-nt.

Pg. 5, Ln 130: “dissolved in 1 dM” should be written as “dissolved in 0,1 M” or dissolved in 100 mM for simplicity.

Pg. 5, lines 151 and 152: “The mouse genes” should be corrected to “The mouse cDNAs”

We have carefully addressed these points in the revised manuscript. We sincerely thank the expert reviewer once again for their thorough evaluation and constructive comments, which have greatly improved the clarity and quality of our article.

Reviewer #2 (Remarks to the Author):

Employing a synthetic DNA substrate containing a site-specific acetaldehyde interstrand crosslink (AA-ICL) located within a stalled replication mimic, the authors performed enzymatic assays with SLX4-XPF-ERCC1 (SXE) to elucidate the incision mechanism employed by this nuclease complex. First, the formation and stability of the AA-ICL substrate was investigated, revealing both slow generation and slow degradation. Second, the molecular products and efficiency of the SXE nuclease was determined on non-crosslinked and crosslinked synthetic AA-ICL and abasic site (Ap)-ICL substrates with varying positioned fluorescent labels. Third, comparison analysis indicated a wide range of DNA substrates for the SXE nuclease complex, whereas the Neil3 glycosylase, responsible for Ap-ICL repair, does not cleave AA-ICLs, underscoring a functional distinction between the two repair mechanisms. Overall, the study represents a straightforward and generally well executed collection of work, offering fundamental insights into biochemical repair of ICLs. For consideration:

The authors appear to suggest that the reactions of ICL substrate formation or degradation do not go to completion (Fig. 2). Has this been attempted, i.e., have the reactions been carried out for months, instead of days? Do the authors have an explanation for why the reactions may not go to completion (is it possible a percentage of the substrate cannot react for some reason)? That could be explored further in the Discussion.

We thank the expert reviewer for their positive assessment of our work and for their constructive comments. We apologise if our interpretation was not clearly conveyed.

Our understanding of the underlying chemistry of Schiff-base formation and hydrolysis (in both AA-ICL and Ap-ICL crosslinks) is that the reaction between the aldehyde and nucleophile reaches an equilibrium, where the rate of crosslink formation equals the rate of hydrolysis. Consequently, the reaction does not go to completion, and a fraction of the substrate remains either unreacted or reversibly crosslinked.

We did not extend the reaction time beyond several days due to concerns about oligonucleotide and fluorescent probe stability, as well as the following rationale: kinetic measurements are most accurately made by assessing initial reaction rates, which capture enzyme activity before substrate depletion, product inhibition, or enzyme instability influence the results. This approach provides a clearer understanding of catalytic efficiency and substrate specificity under near-ideal, linear conditions.

We have now added this clarification regarding the chemical equilibrium in the discussion of the revised manuscript (page 17, line 21-23).

“Acetaldehyde typically binds to intracellular macromolecules by forming imines, or Schiff bases, these adducts are in dynamic equilibrium with their unreacted components and are prone to hydrolysis under physiological conditions (38,39)”

It's unclear whether the authors have compared SXE versus XE incision patterns and efficiency on the full range of substrates. Since prior work has argued for a stimulatory role for SLX4, analysis of this sort might prove insightful.

We thank the reviewer for this thoughtful comment. In our previous work (Hodskinson et al., 2014), we demonstrated that SLX4 stimulates the activity of the XPF–ERCC1 (XE) complex irrespective of the presence of a DNA crosslink — an observation that formed a key part of that study. In the current work, Supplementary Figure 1C provides an updated comparison between SXE and XE on the ICL3 substrate, further confirming that SLX4 enhances incision activity.

Building on these findings, the present study focuses specifically on the activity of the reconstituted SXE complex on a panel of AA-ICL substrates and Ap-ICL lesions. While we did not systematically compare SXE and XE across all substrates in this study. However, our data indicate that the incision rates on crosslinked and non-crosslinked substrates are comparable under the tested conditions. This suggests that SLX4 promotes substrate engagement and catalysis in both contexts. We have now clarified this point in the revised manuscript.

Now we state in the manuscript (page 10, line 27-30):

"Additionally, Figure S1C shows that the SXE complex displays markedly increased activity compared to the XE complex when cleaving the AA-ICL3 substrate, indicating that SLX4 enhances the endonuclease function. This is consistent with our previous findings demonstrating that SLX4 promotes the activity of the XE complex (21)."

Inclusion of a Table that qualitatively summarizes the relative incision efficiencies of SXE versus NEIL3 would be a nice and easy to follow reference.

We thank the reviewer for this thoughtful suggestion and apologise for the omission in our original submission. In response, we have now quantified the relevant gel data and, building on our previous work with NEIL3 activity on Ap-ICL substrates, we have included a new summary table in the revised manuscript. This table qualitatively compares the relative incision efficiencies of the SXE complex and NEIL3 on AA-ICL and Ap-ICL substrates. To aid interpretation, we use intuitive visual symbols, and the accompanying legend clearly states that the table is based on a qualitative assessment of gel band intensities and established enzymatic activities.

We now include this table in the main text (page 16, line 30-34):

Table 1 - Relative incision efficiencies of SXE versus NEIL3
Relative qualitative incision efficiency, based on gel band intensities and known enzymatic activity. Note: Neil3 only cleaves Ap-ICLs when the abasic (Ap) site is adjacent to a single-stranded DNA region on the its 3'-side.

Substrate /Enzyme	SXE	NEIL3
AA-ICL	✓ High incision (++)	☐ No incision
Ap-ICL	✓ High incision (++)	✓ Very high incision (++++)

Reviewer #3 (Remarks to the Author):

Havlikova et al investigates the enzymatic activity of the Slx4-Xpf-Ercc1 (SXE) nuclease complex on synthetic acetaldehyde-induced DNA interstrand crosslinks (AA-ICLs). The authors synthesise site-specific AA-ICLs and demonstrate that SXE can cleave these substrates through dual incisions flanking the crosslink. They also show that SXE cleaves abasic site interstrand crosslinks (Ap-ICLs) but that Neil3 glycosylase does not process AA-ICLs. The experimental work is technically competent, and some very nice biochemical results are shown, but could do with a little toning down of the biological significance.

The main findings of the paper are well supported, that a purified SXE complex can cleave adjacent to an AA-ICL, but that interestingly NEIL3 glycosylase cannot. This demonstrates that like some other ICL agents (eg platinum and nitrogen mustards) only the FA pathway can unhook the damage. The figures as presented are excellent, and well controlled, and the experimental tools developed will be useful to the field of DNA repair research.

However, the contribution of SLX4 is only referenced via previous work – and SLX1 is omitted entirely from the experiments (likely because the SLX4 utilised is a truncated form that doesn't bind SLX1). It is highly likely that XPF:ERCC1 would perform exactly the same cleavage as the SXE proteins used here, based on previous publications

showing it can cleave branched DNA with same cleavage profile as presented (on other structures, but not AA-ICL). Do the authors have results comparing XE and SXE activity in at least one experiment; showing the contribution of SLX4 (if any) would improve the significance of the work.

We are delighted that the expert reviewer finds our work of value, and we thank them for their thoughtful evaluation and constructive comments.

Indeed, we have not included data on the SLX1 nuclease, as current evidence suggests that SLX1 acts primarily in concert with MUS81-EME1 to resolve Holliday junctions (Castor et al., 2013; Gaur et al., 2015; Wyatt et al.). The SLX1–SLX4 complex appears to favour Holliday junction resolution. Furthermore, Kim et al. (2013) demonstrated that a truncated form of SLX4 lacking the SLX1-interacting domain is still capable of rescuing sensitivity to the ICL-inducing agent mitomycin C (MMC), further indicating that SLX1 is dispensable in this context.

Regarding the comparison between the XE and SXE complexes: in our previous work (Hodskinson et al., 2014), we demonstrated that SLX4 strongly enhances the activity of the XPF–ERCC1 (XE) complex, independent of the presence of a DNA crosslink.

In the current work, Figure S1C shows that the SXE complex displays markedly increased activity compared to XE when cleaving the AA-ICL3 substrate, indicating that SLX4 contributes to the enhanced incision. This is consistent with our previous findings on the role of SLX4 in stimulating XE activity (Hodskinson et al., 2014).

Now we state in the manuscript (page 10, line 27-30):

"Additionally, Figure S1C shows that the SXE complex displays markedly increased activity compared to the XE complex when cleaving the AA-ICL3 substrate, indicating that SLX4 enhances the endonuclease function. This is consistent with our previous findings demonstrating that SLX4 promotes the activity of the XE complex (21)."

Moreover, the authors have not provided evidence that SXE is actually responsible for repairing these crosslinks in cells - only that it can cleave them in vitro. As such, it is not responsible to make major conclusions about the FA pathway in response to ICLs generated by "alcohol" when no data exists that directly link the pathway to AA-ICL repair (as opposed to some protein:DNA or other DNA adduct).

The introduction and discussion should be toned down and a clearer distinction should be drawn between what has been demonstrated experimentally versus what is speculated. The discussion overstates the implications of the findings for understanding alcohol-related disease.

Otherwise, a nice strong biochemical paper!

We apologise for the lack of clarity and that the reviewer finds the tone inappropriate. We took extra care to make changes to the introduction and discussions to bring a clear distinction between speculation and experimental evidence. Moderated overall text more tempered and less overstated. We added words underscoring this, like we speculate etc.

To name a few; we changed this (page 18, line 32-34):

"Our findings demonstrate that SXE not only excises alcohol-derived AA-ICLs but also cleaves spontaneously formed Ap-ICLs in a comparable manner, highlighting its role as a versatile FA nuclease."

To this (page 18, line 32-34):

"Our in vitro findings demonstrate that SXE is capable of cleaving alcohol-derived AA-ICLs and Ap-ICLs in a comparable manner, suggesting its potential role as a versatile FA nuclease. However, further studies are needed to confirm its involvement in AA-ICL repair in cells."

Also, we have added this sentence (page 18, line 24-25):

"However, the specific contribution of SXE to ICL repair in vivo under these conditions requires further investigation."

While removing (page 18, line 24): *...,*" and even highlight the critical role of SXE in the FA pathway and ICL repair."

Additionally, there is sufficient biological evidence supporting the involvement of Slx4 and the Fanconi anaemia (FA) pathway in the repair of acetaldehyde-induced ICLs. In previous work, increased sensitivity of SLX4-knockout cells to acetaldehyde was demonstrated, indicating the importance of SLX4 in AA-ICL repair (Figure 2, Garaycochea et al., 2012). In our own prior study, conducted in the laboratories of KJ Patel and Puck Knipscheer, we showed that AA-ICL repair in cells is primarily mediated by the FA pathway (Hoskinson et al., 2020). Finally, in the present study, we show that the AA-ICL is cleaved by the SXE complex in vitro, comprising Slx4, Xpf, and Ercc1, all of which are components of the FA pathway. This strongly suggests SXE's mechanistic role in AA-ICL repair of the FA pathway.